# Fast and Accurate Least-Mean-Squares Solvers

**Alaa Maalouf** *
Alaamalouf12@gmail.com

**Ibrahim Jubran**[*]
ibrahim.jub@gmail.com

**Dan Feldman**
dannyf.post@gmail.com

The Robotics and Big Data Lab,
Department of Computer Science,
University of Haifa,
Haifa, Israel

## Abstract

Least-mean squares (LMS) solvers such as Linear / Ridge / Lasso-Regression, SVD and Elastic-Net not only solve fundamental machine learning problems, but are also the building blocks in a variety of other methods, such as decision trees and matrix factorizations.

We suggest an algorithm that gets a finite set of $n$ $d$-dimensional real vectors and returns a weighted subset of $d + 1$ vectors whose sum is *exactly* the same. The proof in Caratheodory's Theorem (1907) computes such a subset in $O(n^2 d^2)$ time and thus not used in practice. Our algorithm computes this subset in $O(nd)$ time, using $O(\log n)$ calls to Caratheodory's construction on small but "smart" subsets. This is based on a novel paradigm of fusion between different data summarization techniques, known as sketches and coresets.

As an example application, we show how it can be used to boost the performance of existing LMS solvers, such as those in scikit-learn library, up to x100. Generalization for streaming and distributed (big) data is trivial. Extensive experimental results and complete open source code are also provided.

## 1 Introduction and Motivation

Least-Mean-Squares (LMS) solvers are the family of fundamental optimization problems in machine learning and statistics that include linear regression, Principle Component Analysis (PCA), Singular Value Decomposition (SVD), Lasso and Ridge regression, Elastic net, and many more [17, 20, 19, 38, 43, 39, 37]. See formal definition below. First closed form solutions for problems such as linear regression were published by e.g. Pearson [33] around 1900 but were probably known before. Nevertheless, today they are still used extensively as building blocks in both academy and industry for normalization [27, 23, 3], spectral clustering [34], graph theory [42], prediction [11, 36], dimensionality reduction [26], feature selection [16] and many more; see more examples in [18].

Least-Mean-Squares solver in this paper is an optimization problem that gets as input an $n \times d$ real matrix $A$, and another $n$-dimensional real vector $b$ (possibly the zero vector). It aims to minimize the sum of squared distances from the rows (points) of $A$ to some hyperplane that is represented by its normal or vector of $d$ coefficients $x$, that is constrained to be in a given set $X \subseteq \mathbb{R}^d$:

$$\min_{x \in X} f(\|Ax - b\|_2) + g(x). \tag{1}$$

Here, $g$ is called a *regularization term*. For example: in linear regression $X = \mathbb{R}^d$, $f(y) = y^2$ for every $y \in \mathbb{R}$ and $g(x) = 0$ for every $x \in X$. In Lasso $f(y) = y^2$ for every $y \in \mathbb{R}$ and $g(x) = \alpha \cdot \|x\|_1$

---

[*]These authors contributed equally to this work.

for every $x \in \mathbb{R}^d$ and $\alpha > 0$. Such LMS solvers can be computed via the covariance matrix $A^T A$. For example, the solution to linear regression of minimizing $\|Ax - b\|_2$ is $(A^T A)^{-1} A^T b$.

## 1.1 Related work

While there are many LMS solvers and corresponding implementations, there is always a trade-off between their accuracy and running time; see comparison table in [5] with references therein. The reason is related to the fact that computing the covariance matrix of $A$ can be done essentially in one of two ways: (i) summing the $d \times d$ outer product $a_i a_i^T$ of the $i$th row $a_i^T$ of $A$ over every $i$, $1 \leq i \leq n$. This is due to the fact that $A^T A = \sum_{i=1}^{n} a_i a_i^T$, or (ii) factorization of $A$, e.g. using SVD or the QR decomposition [17].

**Numerical issues.** Method (i) is easy to implement for streaming rows of $A$ by maintaining only $d^2$ entries of the covariance matrix for the $n$ vectors seen so far, or maintaining its inverse $(A^T A)^{-1}$ as explained e.g. in [18]. This takes $O(d^2)$ time for each vector insertion and requires $O(d^2)$ memory, which is the same as the desired output covariance matrix. However, every such addition may introduce another numerical error which accumulates over time. This error increases significantly when running the algorithms using 32 bit floating point representation, which is common for GPU computations; see Fig. 2v for example. This solution is similar to maintaining the set of $d$ rows of the matrix $DV^T$, where $A = UDV^T$ is the SVD of $A$, which is not a subset of the original input matrix $A$ but has the same covariance matrix $A^T A = V D^2 V$. A common problem is that to compute $(A^T A)^{-1}$, the matrix $A^T A$ must be invertible. This may not be the case due to numerical issues. In algorithms such as Lasso, the input cannot be a covariance matrix, but only a corresponding matrix whose covariance matrix is $A^T A$, that can be computed from the Cholesky decomposition [6] that returns a left triangular matrix $A$ for the given covariance matrix $A^T A$. However, Cholesky decomposition can be applied only on positive-definite matrices, which is not the case even for small numerical errors that are added to $A^T A$. See Section 4 for more details and empirical evidence.

**Running-time issues.** Method (ii) above utilizes factorizations such as SVD, i.e., $A = UDV^T$ to compute the covariance matrix via $A^T A = V D^2 V^T$ or the QR decomposition $A = QR$ to compute $A^T A = R^T Q^T Q R^T = R^T R$. This approach is known to be much more stable. However, it is much more time consuming: while in theory the running time is $O(nd^2)$ as in the first method, the constants that are hidden in the $O(\cdot)$ notation are significantly larger. Moreover, unlike Method (i), it is impossible to compute such factorizations exactly for streaming data [8].

**Caratheodory's Theorem [7]** states that every point contained in the convex hull of $n$ points in $\mathbb{R}^d$ can be represented as a convex combination of a subset of at most $d + 1$ points, which we call the *Caratheodory set*; see Section 2 and Fig. 1. This implies that we can maintain a weighted (scaled) set of $d^2 + 1$ points (rows) whose covariance matrix is the same as $A$, since $(1/n) \sum_i a_i a_i^T$ is the mean of $n$ matrices and thus in the convex hull of their corresponding points in $\mathbb{R}^{(d^2)}$; see Algorithm 2. The fact that we can maintain such a small sized subset of points instead of updating linear combinations of all the $n$ points seen so far, significantly reduces the numerical errors as shown in Fig. 2v. Unfortunately, computing this set from Caratheodory's Theorem takes $O(n^2 d^2)$ or $O(nd^3)$ time via $O(n)$ calls to an LMS solver. This fact makes it non-practical to use in an LMS solvers, as we aim to do in this work, and may explain the lack of software or source code for this algorithm on the web.

**Approximations via Coresets and Sketches.** In the recent decades numerous approximation and data summarization algorithms were suggested to *approximate* the problem in (1); see e.g. [13, 21, 9, 30] and references therein. One possible approach is to compute a small matrix $S$ whose covariance $S^T S$ approximates, in some sense, the covariance matrix $A^T A$ of the input data $A$. The term *coreset* is usually used when $S$ is a weighted (scaled) subset of rows from the $n$ rows of the input matrix. The matrix $S$ is sometimes called a *sketch* if each rows in $S$ is a linear combination of few or all rows in $A$, i.e. $S = WA$ for some matrix $W \in \mathbb{R}^{s \times n}$. However, those coresets and sketches usually yield $(1 + \varepsilon)$-multiplicative approximations for $\|Ax\|_2^2$ by $\|Sx\|_2^2$ where the matrix $S$ is of $(d/\varepsilon)^{O(1)}$ rows and $x$ may be any vector, or the smallest/largest singular vector of $S$ or $A$; see lower bounds in [14]. Moreover, a $(1 + \varepsilon)$-approximation to $\|Ax\|_2^2$ by $\|Sx\|_2^2$ does not guarantee an approximation to the actual entries or eigenvectors of $A$ by $S$ that may be very different.

**Accurately handling big data.** The algorithms in this paper return *accurate* coresets ($\varepsilon = 0$), which is less common in the literature; see [22] for a brief summary. These algorithms can be used

to compute the covariance matrix $A^T A$ via a scaled subset of rows from the input matrix $A$. Such coresets support unbounded stream of input rows using memory that is *sub-linear* in their size, and also support dynamic/distributed data in parallel. This is by the useful merge-and-reduce property of coresets that allow them to handle big data; see details e.g. in [4]. Unlike traditional coresets that pay additional logarithmic multiplicative factors due to the usage of merge-reduce trees and increasing error, the suggested weighted subsets in this paper do not introduce additional error to the resulting compression since they preserve the desired statistics accurately. The actual numerical errors are measured in the experimental results, with analysis that explain the differences.

A main advantage of a coreset over a sketch is that it preserves sparsity of the input rows [15], which usually reduces theoretical running time. Our experiments show, as expected from the analysis, that coresets can also be used to significantly improve the numerical stability of existing algorithms. Another advantage is that the same coreset can be used for parameter tuning over a large set of candidates. In addition to other reasons, this significantly reduces the running time of such algorithms in our experiments; see Section 4.

## 1.2 Our contribution

A natural question that follows from the previous section is: *can we maintain the optimal solution for LMS problems both accurately and fast?* We answer this question affirmably by suggesting:

  (i) the first algorithm that computes the Caratheodory set of $n$ input points in time that is linear in the input $O(nd)$ for asymptotically large $n$. This is by using a novel approach of coreset/skecthes fusion that is explained in the next section; see Algorithm 1 and Theorem 1.

 (ii) an algorithm that maintains a ("coreset") matrix $S \in \mathbb{R}^{(d^2+1) \times d}$ such that: (a) its set of rows is a scaled subset of rows from $A \in \mathbb{R}^{n \times d}$ whose rows are the input points, and (b) the covariance matrices of $S$ and $A$ are the same, i.e., $S^T S = A^T A$; see Algorithm 2 and Theorem 3.2.

(iii) example applications for boosting the performance of *existing* solvers by running them on the matrix $S$ above or its variants for Linear/Ridge/Lasso Regressions and Elastic-net.

(iv) extensive experimental results on synthetic and real-world data for common LMS solvers of Scikit-learn library with either CPython or Intel's distribution. Either the running time or numerical stability is improved up to two orders of magnitude.

 (v) open code [29] for our algorithms that we hope will be used for the many other LMS solvers and future research as suggested in our Conclusion section; see Section 5.

## 1.3 Novel approach: Coresets meet Sketches

As explained in Section 1.1, the covariance matrix $A^T A$ of $A$ itself can be considered as a sketch which is relatively less numerically stable to maintain (especially its inverse, as desired by e.g. linear regression). The Caratheodory set, as in Definition 2.1, that corresponds to the set of outer products of the rows of $A$ is a coreset whose weighted sum yields the covariance matrix $A^T A$. Moreover, it is more numerically stable but takes much more time to compute; see Theorem 2.2.

To this end, we suggest a meta-algorithm that combines these two approaches: sketches and coresets. It may be generalized to other, not-necessarily accurate, $\varepsilon$-coresets and sketches ($\varepsilon > 0$); see Section 5.

**The input** to our meta-algorithm is 1) a set $P$ of $n$ items, 2) an integer $k \in \{1, \cdots, n\}$ where $n$ is highest numerical accuracy but longest running time, and 3) a pair of coreset and sketch construction schemes for the problem at hand.
**The output** is a coreset for the problem whose construction time is faster than the construction time of the given coreset scheme; see Fig. 1.

**Step I:** Compute a balanced partition $\{P_1, \cdots, P_k\}$ of the input set $P$ into $k$ clusters of roughly the same size. While the correctness holds for any such arbitrary partition (e.g. see Algorithm 3.1), to reduce numerical errors – the best is a partition that minimizes the sum of loss with respect to the problem at hand.

**Step II:** Compute a sketch $S_i$ for each cluster $P_i$, where $i \in \{1, \cdots, k\}$, using the input sketch scheme. This step does not return a subset of $P$ as desired, and is usually numerically less stable.

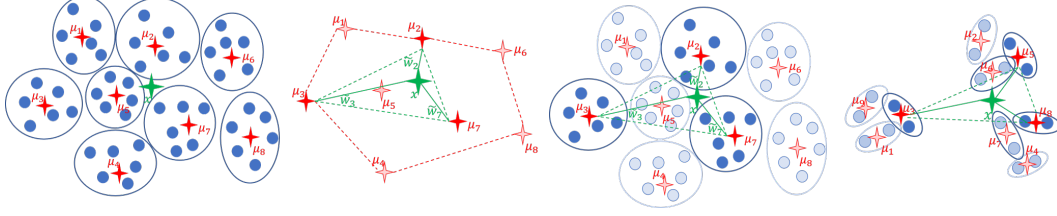

**Figure 1:** *Overview of Algorithm 1 and the steps in Section 1.3. Images left to right: Steps I and II (Partition and sketch steps): A partition of the input weighted set of $n = 48$ points (in blue) into $k = 8$ equal clusters (in circles) whose corresponding means are $\mu_1, \ldots, \mu_8$ (in red). The mean of $P$ (and these means) is $x$ (in green). Step III (Coreset step): Caratheodory (sub)set of $d + 1 = 3$ points (bold red) with corresponding weights (in green) is computed only for these $k = 8 \ll n$ means. Step IV (Recover step): the Caratheodory set is replaced by its corresponding original points (dark blue). The remaining points in $P$ (bright blue) are deleted. Step V (Recursive step): Previous steps are repeated until only $d + 1 = 3$ points remain. This procedure takes $O(\log n)$ iterations for $k = 2d + 2$.*

**Step III:** Compute a coreset $B$ for the union $S = S_1 \cup \cdots \cup S_k$ of sketches from Step II, using the input coreset scheme. Note that $B$ is not a subset (or coreset) of $P$.

**Step IV:** Compute the union $C$ of clusters in $P_1, \cdots, P_k$ that correspond to the selected sketches in Step III, i.e. $C = \bigcup_{S_i \in B} P_i$. By definition, $C$ is a coreset for the problem at hand.

**Step V:** Recursively compute a coreset for $C$ until a sufficiently small coreset is obtained. This step is used to reduce running time, without selecting $k$ that is too small.

We then run an existing solver on the coreset $C$ to obtain a faster accurate solution for $P$. Algorithm 1 and 3.1 are special cases of this meta-algorithm, where the sketch is simply the sum of a set of points/matrices, and the coreset is the existing (slow) implementation of the Caratheodory set from Theorem 2.2.

**Paper organization.** In Section 2 we give our notations, definitions and the current state-of-the-art result. Section 3 presents our main algorithms for efficient computation of the Caratheodory (core-)set and a subset that preserves the inputs covariance matrix, their theorems of correctness and proofs. Section 4 demonstrates the applications of those algorithms to common LMS solvers, with extensive experimental results on both real-world and synthetic data via the Scikit-learn library with either CPython or Intel's Python distributions. We conclude the paper with open problems and future work in Section 5.

## 2  Notation and Preliminaries

For a pair of integers $n, d \geq 1$, we denote by $\mathbb{R}^{n \times d}$ the set of $n \times d$ real matrices, and $[n] = \{1, \cdots, n\}$. To avoid abuse of notation, we use the big $O$ notation where $O(\cdot)$ is a set [12]. A *weighted set* is a pair $(P, u)$ where $P = \{p_1, \cdots, p_n\}$ is an ordered finite set in $\mathbb{R}^d$, and $u : P \to [0, \infty)$ is a positive *weights function*. We sometimes use a matrix notation whose rows contains the elements of $P$ instead of the ordered set notation.

Given a point $q$ inside the convex hull of a set of points $P$, Caratheodory's Theorem proves that there a subset of at most $d + 1$ points in $P$ whose convex hull also contains $q$. This geometric definition can be formulated as follows.

**Definition 2.1** (Caratheodory set). *Let $(P, u)$ be a weighted set of $n$ points in $\mathbb{R}^d$ such that $\sum_{p \in P} u(p) = 1$. A weighted set $(S, w)$ is called a* Caratheodory Set *for $(P, u)$ if: (i) its size is $|S| \leq d + 1$, (ii) its weighted mean is the same, $\sum_{p \in S} w(p) \cdot p = \sum_{p \in P} u(p) \cdot p$, and (iii) its sum of weights is $\sum_{p \in S} w(p) = 1$.*

Caratheodory's Theorem suggests a constructive proof for computing this set in $O(n^2 d^2)$ time [7, 10]; see Algorithm 8 along with an overview and full proof in Section A of the supplementary material [31]. However, as observed e.g. in [32], it can be computed only for the first $m = d + 1$ points, and then be updated point by point in $O(md^2) = O(d^3)$ time per point, to obtain $O(nd^3)$ overall time. This still takes $\Theta(n)$ calls to a linear system solver that returns $x \in \mathbb{R}^d$ satisfying $Ax = b$ for a given matrix $A \in \mathbb{R}^{(d+1) \times d}$ and vector $b \in \mathbb{R}^{d+1}$, in $O(d^3)$ time per call.

**Theorem 2.2** ([7], [32]). *A Caratheodory set $(S, w)$ can be computed for any weighted set $(P, u)$ where $\sum_{p \in P} u(p) = 1$ in $t(n, d) \in O(1) \cdot \min \left\{ n^2 d^2, nd^3 \right\}$ time.*

## 3 Faster Caratheodory Set

In this section, we present our main algorithm that reduces the running time for computing a Caratheodory set from $O(\min\{n^2d^2, nd^3\})$ in Theorem 2.2 to $O(nd)$ for sufficiently large $n$; see Theorem 3.1. A visual illustration of the corresponding Algorithm 1 is shown in Fig. 1. As an application, we present a second algorithm, called CARATHEODORY-MATRIX, which computes a small weighted subset of a the given input that has the same covariance matrix as the input matrix; see Algorithm 2.

**Theorem 3.1** (Caratheodory-Set Booster). *Let $(P, u)$ be a weighted set of $n$ points in $\mathbb{R}^d$ such that $\sum_{p \in P} u(p) = 1$, and $k \geq d + 2$ be an integer. Let $(C, w)$ be the output of a call to* FAST-CARATHEODORY-SET$(P, u, k)$*; See Algorithm 1. Let $t(k, d)$ be the time it takes to compute a Caratheodory Set for $k$ points in $\mathbb{R}^d$, as in Theorem 2.2. Then $(C, w)$ is a Caratheodory set of $(P, u)$ that is computed in time $O\left(nd + t(k, d) \cdot \frac{\log n}{\log(k/d)}\right)$.*

*Proof.* See full proof of Theorem B.1 in the supplementary material [31]. □

**Tuning Algorithm 1 for the fastest running time.** To achieve the fastest running time in Algorithm 1, simple calculations show that when $t(k, d) = kd^3$, i.e., when applying the algorithm from [32], $k = ed$ is the optimal value (that achieves the fastest running time), and when $t(k, d) = k^2d^2$, i.e., when applying the original Caratheodory algorithm (Algorithm 8 in the supplementary material [31]), $k = \sqrt{e}d$ is the value that achieves the fastest running time.

---

**Algorithm 1** FAST-CARATHEODORY-SET$(P, u, k)$; see Theorem 3.1

---

**Input:** A set $P$ of $n$ points in $\mathbb{R}^d$, a (weight) function $u : P \to [0, \infty)$ such that $\sum_{p \in P} u(p) = 1$, and an integer (number of clusters) $k \in \{1, \cdots, n\}$ for the numerical accuracy/speed trade-off.
**Output:** A Caratheodory set of $(P, u)$; see Definition 2.1.

1 $P := P \setminus \{p \in P \mid u(p) = 0\}$.   `// Remove all points with zero weight.`
2 **if** $|P| \leq d + 1$ **then**
3 |   **return** $(P, u)$   `// |P| is already small`
4 $\{P_1, \cdots, P_k\} :=$ a partition of $P$ into $k$ disjoint subsets (clusters), each contains at most $\lceil n/k \rceil$ points.
5 **for** *every* $i \in \{1, \cdots, k\}$ **do**

$$\mu_i := \frac{1}{\sum_{q \in P_i} u(q)} \cdot \sum_{p \in P_i} u(p) \cdot p \quad \text{// the weighted mean of } P_i$$

7 |   $u'(\mu_i) := \sum_{p \in P_i} u(p)$   `// The weight of the ith cluster.`
8 $(\tilde{\mu}, \tilde{w}) :=$ CARATHEODORY$(\{\mu_1, \cdots, \mu_k\}, u')$
    `// see Algorithm 8 in the supplementary material and Theorem 2.2.`
9 $C := \bigcup_{\mu_i \in \tilde{\mu}} P_i$
    `// ` $C$ ` is the union over all clusters ` $P_i \subseteq P$ ` whose representative ` $\mu_i$ ` was chosen for ` $\tilde{\mu}$`.`
10 **for** *every* $\mu_i \in \tilde{\mu}$ *and* $p \in P_i$ **do**
11 |   $w(p) := \dfrac{\tilde{w}(\mu_i)u(p)}{\sum_{q \in P_i} u(q)}$   `// assign weight for each point in ` $C$
12 $(C, w) :=$ FAST-CARATHEODORY-SET$(C, w, k)$   `// recursive call`
13 **return** $(C, w)$

---

**Theorem 3.2.** *Let $A \in \mathbb{R}^{n \times d}$ be a matrix, and $k \geq d^2 + 2$ be an integer. Let $S \in \mathbb{R}^{(d^2+1) \times d}$ be the output of a call to* CARATHEODORY-MATRIX$(A, k)$*; see Algorithm 2. Let $t(k, d)$ be the computation time of* CARATHEODORY *(Algorithm 8) given $k$ points in $\mathbb{R}^d$. Then $A^T A = S^T S$. Furthermore, $S$ is computed in $O\left(nd^2 + t(k, d^2) \cdot \frac{\log n}{\log(k/d^2))}\right)$ time.*

*Proof.* See full proof of Theorem B.2 in the supplementary material [31]. □

**Algorithm 2** CARATHEODORY-MATRIX$(A, k)$; see Theorem 3.2

---

**Input** : A matrix $A = (a_1 \mid \cdots \mid a_n)^T \in \mathbb{R}^{n \times d}$, and an integer $k \in \{1, \cdots, n\}$ for numerical accuracy/speed trade-off.

**Output:** A matrix $S \in \mathbb{R}^{(d^2+1) \times d}$ whose rows are scaled rows from $A$, and $A^T A = S^T S$.

1 **for** *every* $i \in \{1 \cdots, n\}$ **do**

2     Set $p_i \in \mathbb{R}^{(d^2)}$ as the concatenation of the $d^2$ entries of $a_i a_i^T \in \mathbb{R}^{d \times d}$.

     `// The order of entries may be arbitrary but the same for all points.`

3     $u(p_i) := 1/n$

4 $P := \{p_i \mid i \in \{1, \cdots, n\}\}$     `// P is a set of n vectors in ` $\mathbb{R}^{(d^2)}$`.`

5 $(C, w) :=$ FAST-CARATHEODORY-SET$(P, u, k)$ `// C ` $\subseteq$ ` P and ` $|C| = d^2 + 1$ ` by Theorem 3.1`

6 $S :=$ a $(d^2 + 1) \times d$ matrix whose $i$th row is $\sqrt{n \cdot w(p_i)} \cdot a_i^T$ for every $p_i \in C$.

7 **return** $S$

---

| Solver | Objective function | Python's Package | Example Python's solver |
|---|---|---|---|
| Linear regression [6] | $\|Ax - b\|_2^2$ | `scipy.linalg` | `LinearRegression(A, b)` |
| Ridge regression [19] | $\|Ax - b\|_2^2 + \alpha \|x\|_2^2$ | `sklearn.linear_model` | `RidgeCV(A, b, \mathbb{A}, m)` |
| Lasso regression [39] | $\frac{1}{2n}\|Ax - b\|_2^2 + \alpha \|x\|_1$ | `sklearn.linear_model` | `LassoCV(A, b, \mathbb{A}, m)` |
| Elastic-Net regression [43] | $\frac{1}{2n}\|Ax - b\|_2^2 + \rho\alpha \|x\|_1 + \frac{(1-\rho)}{2}\alpha \|x\|_1$ | `sklearn.linear_model` | `ElasticNetCV(A, b, \mathbb{A}, \rho, m)` |

*Table 1:* Four LMS solvers that were tested with Algorithm 3. Each procedure gets a matrix $A \in \mathbb{R}^{n \times d}$, a vector $b \in \mathbb{R}^n$ and aims to compute $x \in \mathbb{R}^d$ that minimizes its objective function. Additional regularization parameters include $\alpha > 0$ and $\rho \in [0, 1]$. The Python's solvers use $m$-fold cross validation over every $\alpha$ in a given set $\mathbb{A} \subseteq [0, \infty)$.

## 4 Experimental Results

In this section we apply our fast construction of the Carathoodory Set $S$ from the previous section to boost the running time of common LMS solvers in Table 1 by a factor of tens to hundreds, or to improve their numerical accuracy by a similar factor to support, e.g., 32 bit floating point representation as in Fig. 2v. This is by running the given solver as a black box on the small matrix $C$ that is returned by Algorithms 4–7, which is based on $S$. That is, our algorithm does not compete with existing solvers but relies on them, which is why we called it "booster". Open code for our algorithms is provided [29].

$m$-**folds cross validation (CV).** We briefly discuss the CV technique which is utilized in common LMS solvers. Given a parameter $m$ and a set of real numbers $\mathbb{A}$, to select the optimal value $\alpha \in \mathbb{A}$ of the regularization term, the existing Python's LMS solvers partition the rows of $A$ into $m$ folds (subsets) and run the solver $m \cdot |\mathbb{A}|$ times, each run is done on a concatenation of $m - 1$ folds (subsets) and $\alpha \in \mathbb{A}$, and its result is tested on the remaining "test fold". Finally, the cross validation returns the parameters that yield the optimal (minimal) value on the test fold; see [25] for details.

**From Caratheodory Matrix to LMS solvers.** As stated in Theorem 3.2, Algorithm 2 gets an input matrix $A \in \mathbb{R}^{n \times d}$ and an integer $k > d + 1$, and returns a matrix $S \in \mathbb{R}^{(d^2+1) \times d}$ of the same covariance $A^T A = S^T S$, where $k$ is a parameter for setting the desired numerical accuracy. To "learn" a given label vector $b \in \mathbb{R}^n$, Algorithm 3 partitions the matrix $A' = (A \mid b)$ into $m$ partitions, computes a subset for each partition that preserves its covariance matrix, and returns the union of subsets as a pair $(C, y)$ where $C \in \mathbb{R}^{(m(d+1)^2+m) \times d}$ and $y \in \mathbb{R}^{m(d+1)^2+m}$. For $m = 1$ and every $x \in \mathbb{R}^d$,

$$\|Ax - b\| = \left\|A'(x \mid -1)^T\right\| = \left\|(C \mid y)(x \mid -1)^T\right\| = \|Cx - y\|, \qquad (2)$$

where the second and third equalities follow from Theorem 3.2 and the construction of $C$, respectively. This enables us to replace the original pair $(A, b)$ by the smaller pair $(C, y)$ for the solvers in Table 1 as in Algorithms 4–7. A scaling factor $\beta$ is also needed in Algorithms 6–7.

To support CV with $m > 1$ folds, Algorithm 3 computes a coreset for each of the $m$ folds (subsets of the data) in Line 4 and concatenates the output coresets in Line 5. Thus, (2) holds similarly for each fold (subset) when $m > 1$.

**The experiments.** We applied our CARATHEODORY-MATRIXcoreset from Algorithm 2 on common Python's SKlearn LMS-solvers that are described in Table 1. Most of these experiments were repeated twice: using the default CPython distribution [40] and Intel's distribution [28] of Python. All the experiments were conducted on a standard Lenovo Z70 laptop with an Intel i7-5500U CPU @ 2.40GHZ and 16GB RAM. We used the 3 following real-world datasets:

(i) 3D Road Network (North Jutland, Denmark) [24]. It contains $n = 434874$ records. We used the $d = 2$ attributes: "Longitude" [Double] and "Latitude" [Double] to predict the attribute "Height in meters" [Double].

(ii) Individual household electric power consumption [1]. It contains $n = 2075259$ records. We used the $d = 2$ attributes: "global active power" [kilowatt - Double], "global reactive power" [kilowatt - Double]) to predict the attribute "voltage" [volt - Double].

(iii) House Sales in King County, USA [2]. It contains $n = 21,600$ records. We used the following $d = 8$ attributes: "bedrooms" [integer], "sqft living" [integer], "sqft lot" [integer], "floors" [integer], "waterfront" [boolean], "sqft above" [integer], "sqft basement" [integer], "year built" [integer]) to predict the "house price" [integer] attribute.

The synthetic data consists of an $n \times d$ matrix $A$ and vector $b$ of length $n$, both of uniform random entries in $[0, 1000]$. As expected by the analysis, since our compression introduces no error to the computation accuracy, the actual values of the data had no affect on the results, unlike the size of the input which affects the computation time. Table 2 summarizes the experimental results.

## 4.1 Competing methods

We now present other sketches for improving the practical running time of LMS solvers; see discussion in Section 4.2.

**SKETCH + CHOLESKY** is a method which simply sums the 1-rank matrices of outer products of rows in the input matrix $A' = (A \mid b)$ which yields its covariance matrix $B = A'^T A'$. The Cholesky decomposition $B = L^T L$ then returns a small matrix $L \in \mathbb{R}^{d \times d}$ that can be plugged to the solvers, similarly to our coreset.

**SKETCH + INVERSE** is applied in the special case of linear regression, where one can avoid applying the Cholesky decomposition and can compute the solution $(A^T A)^{-1} A^T b$ directly after maintaining $A^T A$ and $A^T b$ for the data seen so far.

## 4.2 Discussion

**Running time.** The number of rows in the reduced matrix $C$ is $O(d^2)$, which is usually much smaller than the number $n$ of rows in the original matrix $A$. This also explains why some coresets (dashed red line) failed for small values of $n$ in Fig. 2b,2c,2h and 2i. The construction of $C$ takes $O(nd^2)$. Solving linear regression takes the same time, with or without the coreset. However, the constants hidden in the $O$ notation are much smaller since the time for computing $C$ becomes neglectable for large values of $n$, as shown in Fig. 2u. We emphasize that, unlike common coresets, there is *no accuracy loss* due to the use of our coreset, ignoring $\pm 10^{-15}$ additive errors/improvements. The improvement in running time due to our booster is in order of up to x10 compared to the algorithm's running time on the original data, as shown in Fig. 2m– 2n. The contribution of the coreset is significant, already for smaller values of $n$, when it boosts other solvers that use cross validation for parameter tuning as explained above. In this case, the time complexity reduced by a factor of $m \cdot |\mathbb{A}|$ since the coreset is computed only once for each of the $m$ folds, regardless of the size $|\mathbb{A}|$. In practice, the running time is improved by a factor of x10–x100 as shown for example in Fig. 2a– 2c. As shown in the graphs, e.g., Fig. 2u, the computations via Intel's Python distribution reduced the running times by 15-40% compared to the default CPython distribution, with or without the booster. This is probably due to its tailored implementation for our hardware.

**Numerical stability.** The SKETCH + CHOLESKY method is simple and accurate in theory, and there is no hope to improve its running time via our much more involved booster. However, it is numerically unstable in practice for the reasons that are explained in Section 1.1. In fact, on

| Figure | Algorithm's number | x/y Axes labels | Python Distribution | Dataset | Input Parameter |
|---|---|---|---|---|---|
| 2a,2b,2c | 5–7 | Size/Time for various $d$ | CPython | Synthetic | $m = 3, |\mathbb{A}| = 100$ |
| 2d,2e,2f | 5–7 | Size/Time for various $|\mathbb{A}|$ | CPython | Synthetic | $m = 3, d = 7$ |
| 2g,2h,2i | 5–7 | Size/Time for various $d$ | Intel's | Synthetic | $m = 3, |\mathbb{A}| = 100$ |
| 2j,2k,2l | 5–7 | Size/Time for various $|\mathbb{A}|$ | Intel's | Synthetic | $m = 3, d = 7$ |
| 2m,2n | 5–7 | $|\mathbb{A}|$/Time | CPython | Datasets (i),(ii) | $m = 3, |\mathbb{A}| = 100$ |
| 2o,2p | 5–7 | $|\mathbb{A}|$/Time | Intel's | Datasets (i),(ii) | $m = 3, |\mathbb{A}| = 100$ |
| 2q,2r | 5–7 | Time/maximal $|\mathbb{A}|$ than is feasible | CPython | Datasets (i),(ii) | $m = 3$ |
| 2s,2t | 5–7 | Time/maximal $|\mathbb{A}|$ than is feasible | Intel's | Datasets (i),(ii) | $m = 3$ |
| 2u | 4 | Size/Time for various Distributions | CPython, Intel's | Synthetic | $m = 64, d = 15$ |
| 2v | 4 | Error/Count Histogram + Size/Error | CPython | Datasets (i),(iii) | $m = 1$ |

*Table 2: **Summary of experimental results**. CPython [40] and Intel's [28] distributions were used. The input: $A \in \mathbb{R}^{n \times d}$ and $b \in \mathbb{R}^n$, where $n$ is "Data size". CV used $m$ folds for evaluating each parameter in $\mathbb{A}$. The chosen number of clusters in Algorithm 3 is $k = 2(d + 1)^2 + 2$ in order to have $O(\log n)$ iterations in Algorithm 1, and $\rho = 0.5$ for Algorithm 7. Computation time includes the computation of the reduced input $(C, y)$; See Section 3. The histograms consist of bins along with the number of errors that fall in each bin.*

most of our experiments we could not even apply this technique at all using 32-bit floating point representation. This is because the resulting approximation to $A'^T A'$ was not a positive definite matrix as required by the Cholesky Decomposition, and we could not compute the matrix $L$ at all. In case of success, the running time of the booster was slower by at most a factor of 2 but even in these cases numerical accuracy was improved up to orders of magnitude; See Fig. 2v for histogram of errors using such 32-bit float representation which is especially common in GPUs for saving memory, running time and power [41]. For the special case of linear regression, we can apply SKETCH + INVERSE, which still has large numerical issues compared to our coreset computation as shown in Fig. 2v.

---

**Algorithm 3** LMS-CORESET$(A, b, m, k)$

**Input:** A matrix $A \in \mathbb{R}^{n \times d}$, a vector $b \in \mathbb{R}^n$, a number (integer) $m$ of cross-validation folds, and an integer $k \in \{1, \cdots, n\}$ that denotes accuracy/speed trade-off.

**Output:** A matrix $C \in \mathbb{R}^{O(md^2) \times d}$ whose rows are scaled rows from $A$, and a vector $y \in \mathbb{R}^d$.

1 $A' := (A \mid b)$     // A matrix $A' \in \mathbb{R}^{n \times (d+1)}$
2 $\{A'_1, \cdots, A'_m\} :=$ a partition of the rows of $A'$ into $m$ matrices, each of size $\left(\frac{n}{m}\right) \times (d + 1)$
3 **for** *every* $i \in \{1, \cdots, m\}$ **do**
4 $\quad$ $S_i := $ CARATHEODORY-MATRIX$(A'_i, k)$  // see Algorithm 2
5 $S := (S_1^T | \cdots | S_m^T)^T$ // concatenation of the $m$ matrices into a single matrix of $m(d + 1)^2 + m$ rows and $d + 1$ columns

6 $C :=$ the first $d$ columns of $S$

7 $y :=$ the last column of $S$

8 **return** $(C, y)$

---

**Algorithm 4** LINREG-BOOST$(A, b, m, k)$

1 $(C, y) :=$ LMS-CORESET$(A, b, m, k)$
2 $x^* := $ LinearRegression$(C, y)$
3 **return** $x^*$

**Algorithm 5** RIDGECV-BOOST$(A, b, \mathbb{A}, m, k)$

1 $(C, y) :=$ LMS-CORESET$(A, b, m, k)$
2 $(x, \alpha) := $ RidgeCV$(C, y, \mathbb{A}, m)$
3 **return** $(x, \alpha)$

---

**Algorithm 6** LASSOCV-BOOST$(A, b, \mathbb{A}, m, k)$

1 $(C, y) :=$ LMS-CORESET$(A, b, m, k)$
2 $\beta := \sqrt{\left(m \cdot (d + 1)^2 + m\right)/n}$
3 $(x, \alpha) := $ LassoCV$(\beta \cdot C, \beta \cdot y, \mathbb{A}, m)$
4 **return** $(x, \alpha)$

**Algorithm 7** ELASTICCV-BOOST$(A, b, m, \mathbb{A}, \rho, k)$

1 $(C, y) :=$ LMS-CORESET$(A, b, m, k)$
2 $\beta := \sqrt{\left(m \cdot (d + 1)^2 + m\right)/n}$
3 $(x, \alpha) := $ ElasticNetCV$(\beta \cdot C, \beta \cdot y, \mathbb{A}, \rho, m)$
4 **return** $(x, \alpha)$

---

## 5 Conclusion and Future Work

We presented a novel framework that combines sketches and coresets. As an example application, we proved that the set from the Caratheodory Theorem can be computed in $O(nd)$ overall time for sufficiently large $n$ instead of the $O(n^2 d^2)$ time as in the original theorem. We then generalized the result for a matrix $S$ whose rows are a weighted subset of the input matrix and their covariance matrix is the same. Our experimental results section shows how to significantly boost the numerical

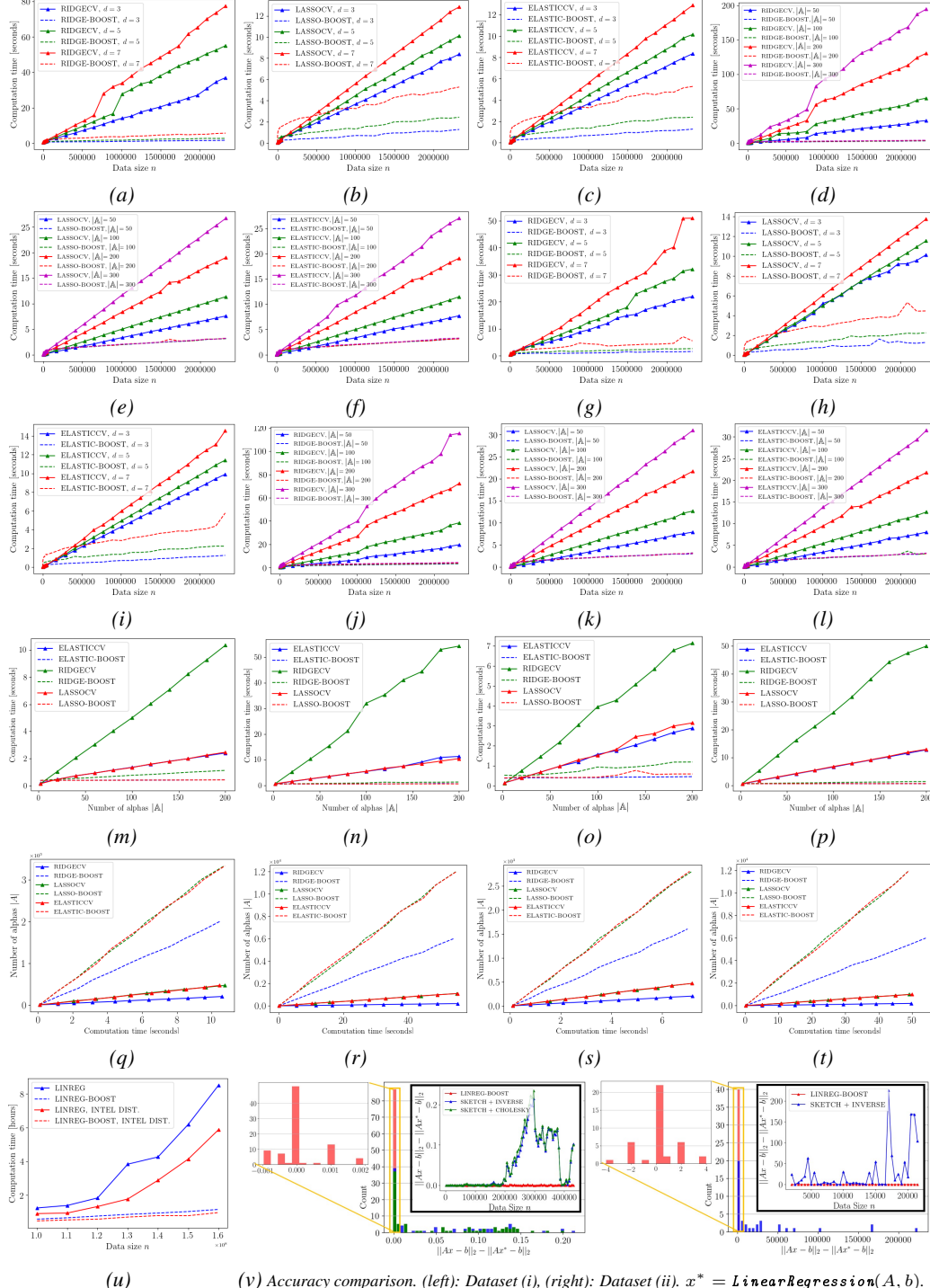

*(a)*  *(b)*  *(c)*  *(d)*

*(e)*  *(f)*  *(g)*  *(h)*

*(i)*  *(j)*  *(k)*  *(l)*

*(m)*  *(n)*  *(o)*  *(p)*

*(q)*  *(r)*  *(s)*  *(t)*

*(u)*  *(v) Accuracy comparison. (left): Dataset (i), (right): Dataset (ii). $x^* = \mathcal{LinearRegression}(A, b)$. $x$ was computed using the methods specified in the legend; see Section 4.2.*

*Figure 2: Experimental results; see Table 2.*

stability or running time of existing LMS solvers by applying them on $S$. Future work includes: (a) applications of our framework to combine other sketch-coreset pairs e.g. as listed in [35], (b) Experiments for streaming/distributed/GPU data, and (c) experiments with higher dimensional data: we may compute each of the $O(d^2)$ entries in the covariance matrix by calling our algorithm with $d = 2$ and the corresponding pair of columns in the $d$ columns of the input matrix.

# 6 acknowledgements

We thank Rafi Dalla-Torre and Benjamin Lastmann from Samsung Research Israel for the fruitful debates and their useful review of our code.

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
