[Supplementary Material]

## A   Slow Caratheodory Implementation

---

**Algorithm 8** CARATHEODORY$(P, u)$

---

**Input** : A weighted set $(P, u)$ of $n$ points in $\mathbb{R}^d$.
**Output:** A Caratheodory set $(S, w)$ for $(P, u)$ in $O(n^2 d^2)$ time.

1 **if** $n \leq d + 1$ **then**
2 $\quad$ **return** $(P, u)$
3 Identify $P = \{p_1, \cdots, p_n\}$
4 **for** *every* $i \in \{2, \cdots, n\}$ **do**
5 $\quad a_i := p_i - p_1$
6 $A := (a_2 \mid \cdots \mid a_n)$ // $A \in \mathbb{R}^{d \times (n-1)}$
7 Compute $v = (v_2, \cdots, v_n)^T \neq 0$ such that $Av = 0$.

8 $v_1 := -\displaystyle\sum_{i=2}^{n} v_i$

9 $\alpha := \min \left\{ \dfrac{u_i}{v_i} \mid i \in \{1, \cdots, n\} \text{ and } v_i > 0 \right\}$

10 $w_i := u_i - \alpha v_i$ for every $i \in \{1, \cdots, n\}$.

11 $S := \{p_i \mid w_i > 0 \text{ and } i \in \{1, \cdots, n\}\}$
$\quad$ **if** $|S| > d + 1$ **then**
12 $\quad (S, w) := $ CARATHEODORY$(S, w)$
13 **return** $(S, w)$

---

**Overview of Algorithm 8 and its correctness.** The input is a weighted set $(P, u)$ whose points are denoted by $P = \{p_1, \cdots, p_n\}$. We assume $n > d + 1$, otherwise $(S, w) = (P, u)$ is the desired coreset. Hence, the $n - 1 > d$ points $p_2 - p_1, p_3 - p_1, \ldots, p_n - p_1 \in \mathbb{R}^d$ must be linearly dependent. This implies that there are reals $v_2, \cdots, v_n$, which are not all zeros, such that

$$\sum_{i=2}^{n} v_i (p_i - p_1) = 0. \tag{2}$$

These reals are computed in Line 7 by solving system of linear equations. This step dominates the running time of the algorithm and takes $O(nd^2)$ time using e.g. SVD. The definition

$$v_1 = -\sum_{i=2}^{n} v_i \tag{3}$$

in Line 8, guarantees that

$$v_j < 0 \text{ for some } j \in [n], \tag{4}$$

and that

$$\sum_{i=1}^{n} v_i p_i = v_1 p_1 + \sum_{i=2}^{n} v_i p_i = \left( -\sum_{i=2}^{n} v_i \right) p_1 + \sum_{i=2}^{n} v_i p_i = \sum_{i=2}^{n} v_i (p_i - p_1) = 0, \tag{5}$$

where the second equality is by (3), and the last is by (2). Hence, for every $\alpha \in \mathbb{R}$, the weighted mean of $P$ is

$$\sum_{i=1}^{n} u_i p_i = \sum_{i=1}^{n} u_i p_i - \alpha \sum_{i=1}^{n} v_i p_i = \sum_{i=1}^{n} (u_i - \alpha v_i) p_i, \tag{6}$$

where the first equality holds since $\sum_{i=1}^{n} v_i p_i = 0$ by (5). The definition of $\alpha$ in Line 9 guarantees that $\alpha v_{i^*} = u_{i^*}$ for some $i^* \in [n]$, and that $u_i - \alpha v_i \geq 0$ for every $i \in [n]$. Hence, the set $S$ that is defined in Line 11 contains at most $n - 1$ points, and its set of weights $\{u_i - \alpha v_i\}$ is non-negative. Notice that if $\alpha = 0$, we have that $w_j = u_j > 0$ for some $j \in [n]$. Otherwise, if $\alpha > 0$, by (4) there is $j \in [n]$ such that $v_j < 0$, which yields that $w_j = u_j - \alpha v_j > 0$. Hence, in both cases there is $w_j > 0$ for some $j \in [n]$. Therefore, $|S| \neq \emptyset$.

The sum of the positive weights is thus the total sum of weights,

$$\sum_{p_i \in S} w_i = \sum_{i=1}^{n}(u_i - \alpha v_i) = \sum_{i=1}^{n} u_i - \alpha \cdot \sum_{i=1}^{n} v_i = 1,$$

where the last equality hold by (3), and since $u$ sums to 1. This and (6) proves that $(S, w)$ is a Caratheodory set of size $n - 1$ for $(P, u)$; see Definition 2.1. In Line 12 we repeat this process recursively until there are at most $d + 1$ points left in $S$. For $O(n)$ iterations, the overall time is thus $O(n^2 d^2)$.

## B  Faster Caratheodory Set

**Theorem B.1** (Theorem 3.1). *Let $(P, u)$ be a weighted set of $n$ points in $\mathbb{R}^d$ such that $\sum_{p \in P} u(p) = 1$, and $k \geq d + 2$ be an integer. Let $(C, w)$ be the output of a call to* FAST-CARATHEODORY-SET$(P, u, k)$*; See Algorithm 1. Let $t(k, d)$ be the time it takes to compute a Caratheodory Set for $k$ points in $\mathbb{R}^d$, as in Theorem 2.2. Then $(C, w)$ is a Caratheodory set of $(P, u)$ that is computed in time*

$$O\left(nd + t(k, d) \cdot \frac{\log n}{\log(k/d)}\right).$$

*Proof.* We use the notation and variable names as defined in Algorithm 1 from Section 3.

First, at Line 1 we remove all the points in $P$ which have zero weight, since they do not contribute to the weighted sum. Therefore, we now assume that $u(p) > 0$ for every $p \in P$ and that $|P| = n$. Identify the input set $P = \{p_1, \cdots, p_n\}$ and the set $C$ that is computed at Line 9 of Algorithm 1 as $C = \{c_1, \cdots, c_{|C|}\}$. We will first prove that the weighted set $(C, w)$ that is computed in Lines 9–11 at an arbitrary iteration is a Caratheodory set for $(P, u)$, i.e., $\sum_{p \in P} u(p) \cdot p = \sum_{p \in C} w(p) \cdot p$, $\sum_{p \in P} u(p) = \sum_{p \in C} w(p)$ and $|C| \leq (d + 1) \cdot \left\lceil \frac{n}{k} \right\rceil$.

Let $(\tilde{\mu}, \tilde{w})$ be the pair that is computed during the execution the current iteration at Line 8. By Theorem 2.2 and Algorithm 8, the pair $(\tilde{\mu}, \tilde{w})$ is a Caratheodory set of the weighted set $(\{\mu_1, \cdots, \mu_k\}, u')$. Hence,

$$\sum_{\mu_i \in \tilde{\mu}} \tilde{w}(\mu_i) = 1, \quad \sum_{\mu_i \in \tilde{\mu}} \tilde{w}(\mu_i)\mu_i = \sum_{i=1}^{k} u'(\mu_i) \cdot \mu_i \text{ and } |\tilde{\mu}| \leq d + 1. \tag{7}$$

By the definition of $\mu_i$, for every $i \in \{1, \cdots, k\}$

$$\sum_{i=1}^{k} u'(\mu_i) \cdot \mu_i = \sum_{i=1}^{k} u'(\mu_i) \cdot \left(\frac{1}{u'(\mu_i)} \cdot \sum_{p \in P_i} u(p) \cdot p\right) = \sum_{i=1}^{k} \sum_{p \in P_i} u(p)p = \sum_{p \in P} u(p)p. \tag{8}$$

We now have that

$$\sum_{p \in C} w(p)p = \sum_{\mu_i \in \tilde{\mu}} \sum_{p \in P_i} \frac{\tilde{w}(\mu_i)u(p)}{u'(\mu_i)} \cdot p = \sum_{\mu_i \in \tilde{\mu}} \tilde{w}(\mu_i) \sum_{p \in P_i} \frac{u(p)}{u'(\mu_i)}p = \sum_{\mu_i \in \tilde{\mu}} \tilde{w}(\mu_i)\mu_i$$
$$= \sum_{i=1}^{k} u'(\mu_i) \cdot \mu_i = \sum_{p \in P} u(p)p, \tag{9}$$

where the first equality holds by the definitions of $C$ and $w$, the third equality holds by the definition of $\mu_i$ at Line 5, the fourth equality is by (7), and the last equality is by (8).

The new sum of weights is equal to

$$\sum_{p \in C} w(p) = \sum_{\mu_i \in \tilde{\mu}} \sum_{p \in P_i} \frac{\tilde{w}(\mu_i)u(p)}{u'(\mu_i)} = \sum_{\mu_i \in \tilde{\mu}} \frac{\tilde{w}(\mu_i)}{u'(\mu_i)} \cdot \sum_{p \in P_i} u(p) = \sum_{\mu_i \in \tilde{\mu}} \frac{\tilde{w}(\mu_i)}{u'(\mu_i)} \cdot u'(\mu_i) = \sum_{\mu_i \in \tilde{\mu}} \tilde{w}(\mu_i) = 1,$$
$$\tag{10}$$

where the last equality is by (7).

Combining (9) and (10) yields that the weighted $(C, w)$ computed before the recursive call at Line 13 of the algorithm is a Caratheodory set for the weighted input set $(P, u)$. Since at each iteration we either return such a Caratheodory set $(C, w)$ at Line 13 or return the input weighted set $(P, u)$ itself at Line 3, by induction we conclude that the output weighted set of a call to FAST-CARATHEODORY-SET$(P, u, k)$ is a Caratheodory set for the original input $(P, u)$.

By (7) we have that $C$ contains at most $(d + 1)$ clusters from $P$ and at most $|C| \leq (d + 1) \cdot \left\lceil \frac{n}{k} \right\rceil$ points. Hence, there are at most $\log_{\frac{k}{d+1}}(n)$ recursive calls before the stopping condition in line 2 is satisfied. The time complexity of each iteration is $n' + t(k, d)$ where $n' = |P| \cdot d$ is the number of points in the current iteration. Thus the total running time of Algorithm 1 is

$$\sum_{i=1}^{\log_{\frac{k}{d+1}}(n)} \left( \frac{nd}{2^{i-1}} + t(k, d) \right) \leq 2nd + \log_{\frac{k}{d+1}}(n) \cdot t(k, d) \in O\left( nd + \frac{\log n}{\log(k/(d+1))} \cdot t(k, d) \right).$$

$\square$

**Theorem B.2** (Theorem 3.2). *Let $A \in \mathbb{R}^{n \times d}$ be a matrix, and $k \geq d^2 + 2$ be an integer. Let $S \in \mathbb{R}^{(d^2+1) \times d}$ be the output of a call to CARATHEODORY-MATRIX$(A, k)$; see Algorithm 2. Let $t(k, d)$ be the computation time of CARATHEODORY given $k$ point in $\mathbb{R}^{d^2}$. Then $S$ satisfies that $A^T A = S^T S$. Furthermore, $S$ can be computed in $O(nd^2 + t(k, d^2) \cdot \frac{\log n}{\log(k/d^2)})$ time.*

*Proof.* We use the notation and variable names as defined in Algorithm 2 from Section 3.

Since $(C, w)$ at Line 5 of Algorithm 2 is the output of a call to FAST-CARATHEODORY-SET$(P, u, k)$, by Theorem 3.1 we have that: (i) the weighted means of $(C, w)$ and $(P, u)$ are equal, i.e.,

$$\sum_{p \in P} u(p) \cdot p = \sum_{p \in C} w(p) \cdot p, \tag{11}$$

(ii) $|C| \leq d^2 + 1$ since $P \subseteq \mathbb{R}^{(d^2)}$, and (iii) $C$ is computed in $O(nd^2 + \log_{\frac{k}{d^2+1}}(n) \cdot t(k, d^2))$ time.

Combining (11) with the fact that $p_i$ is simply the concatenation of the entries of $a_i a_i^T$, we have that

$$\sum_{p_i \in P} u(p_i) a_i a_i^T = \sum_{p_i \in C} w(p_i) \cdot a_i a_i^T. \tag{12}$$

By the definition of $S$ in Line 6, we have that

$$S^T S = \sum_{p_i \in C} (\sqrt{n \cdot w(p_i)} \cdot a_i)(\sqrt{n \cdot w(p_i)} \cdot a_i)^T = n \cdot \sum_{p_i \in C} w(p_i) \cdot a_i a_i^T. \tag{13}$$

We also have that

$$A^T A = \sum_{i=1}^n a_i a_i^T = n \cdot \sum_{p_i \in P} (1/n) a_i a_i^T = n \cdot \sum_{p_i \in P} u(p_i) a_i a_i^T, \tag{14}$$

where the second derivation holds since $u \equiv 1/n$. Theorem 3.2 now holds by combining (12), (13) and (14) as

$$S^T S = n \cdot \sum_{p_i \in C} w(p_i) \cdot a_i a_i^T = n \cdot \sum_{p_i \in P} u(p_i) a_i a_i^T = A^T A.$$

**Running time:** Computing the weighted set $(P, u)$ at Lines 1–4 takes $O(nd^2)$ time, since it takes $O(d^2)$ time to compute each of the $n$ points in $P$.

By Theorem 3.1, Line 5 takes $O(nd^2 + t(k, d^2) \cdot \frac{\log n}{\log(k/d^2)})$ to compute a CARATHEODORY for the the weighted set $(P, u)$, and finally Line 6 takes $O(d^3)$ for building the matrix $S$. Hence, the overall running time of Algorithm 2 is $O(nd^2 + t(k, d^2) \cdot \frac{\log n}{\log(k/d^2)})$. $\square$