[Reviews · NeurIPS 2019]

Reviewer 1



To the best of my knowledge, the paper is the first to introduce an *exact* coreset/sketch construction for linear regression problems (i.e. there is no loss in quality by using the coreset/sketch instead of the original input data). In contrast, other work, e.g. [13] (see paper), focuses on coresets/sketches for which one can only obtain approximation guarantees. To achieve this result, the paper combines known results, but in a non-trivial non-obvious way. The paper gives proves for all theorems (in the appendix); these proofs are easy to follow. Also the overview on the contributions, related work, and the high-level overview on the approach are useful. In the experiments, the contribution of the paper helps to speedup known LMS solvers in the scikit-learn library by a large factors. Hence, the contribution of the paper adds a real measurable value for practical applications. However, sometimes the descriptons are a bit hard to follow and non-straightforward. Especially the "weighted set" vs. matrix notation is a bit hard to follow: - The definition "P={p_1,..,p_n} is an ordered finite set" is strange. The given set might very well be a multiset (and ordered), so a tuple notation like (p_1,...,p_n) might be clearer. - In Algorithm 3, "C in R^{...} a weighted subset of rows in A", matrices are interpreted as sets. How about "rows of C ... are linear combinations of rows in A" (to avoid the "ordered multi-set" problem alltogether)? - In Algorithm 2, "S in R^{...} whose union of rows is a weighted subset of A". It's not a weighted subset. Its rows coincide with a certain subset of rows of A that have been scaled. (For people that are used to coresets/sketches in the context of clustering this can be very confusing). Some more minor remarks: - [8] is mentioned Theorem 8.8 directly, while [27] is mentioned separately in the text twice - but it seems [27] explains Theorem 2.2 as well? - "Then (C,u) is a Caratheodory set of (P,w) that can be computed in time..." - The given algorithm has this running time, not some other algorithm that can compute (C,u) - wikipedia as a reference... - In l. 95, "linear in the input O(nd) for asymptotically large n" is a weird way of ignoring the dependency on d, better write "linear in n and polynomial in d". Moreover, "only log(n) calls to an LMS solver" is not so meaningful because it's unclear to what the LMS solver is applied. - I think it's nice that the paper does use O-notation correctly but the explicit remark in l. 141 seems unnecessary since it's only about one "\in" instead of "=" (and all the theorems the use abbreviations anyway ("comuted in time O(...)" = {...}=set) - The font size used in the figures in Figure 2 are way too tiny. The results shown in Figures 2(r) and 2(s) are nice (if you can zomm ;)) - but it's aweful how the sub-figures have been pasted together (and again, how tiny the fonts are) - l. 212 "time for .. becomes neglected" - l. 215 "is much significant" Some typos: - l. 43: "this errors increases" - l. 178 "from [the] previous section" - l. 68 "on the web T" - l. 199 "the common Python's solvers" - l. 130 "a sufficiently coreset"

Reviewer 2



This is a groundbreaking result in the area of sketch and coreset. A large number of existing algorithms in sketch and coreset are randomized and approximate, which prevents those algorithms from being used in practice. The current paper presents a new deterministic coreset which receives a matrix A with n rows and d columns, and outputs a matrix A' with d^2 + 1 rows and d columns, such that rows of A' are scaled rows of A and A^TA equals A'^TA' *with no approximation*. Caratheodory's Theorem in fact implies the existence of such matrix A' and an inefficient algorithm, and the main contribution of this paper is a new practical algorithm which calculates the Caratheodory set with much faster running time. The new algorithm is elegant yet very interesting. Technically, the algorithm first partitions the given data points into k clusters and then calls the slow algorithm on k data points which correspond to the weighted mean of these k clusters. Using the output of the slow algorithm, by appropriately reweighting the data points, one can reduce the number of data points from n to O(n*k/d), at which point one can recursively call the algorithm. Thus, the algorithm terminates after log n / log (k / d) recursive calls. Another advantage of this paper is its strong experiment section. The discussion and plots presented in this section clearly demonstrate the practicality of the proposed coreset. In summary, this is a strong paper with a novel coreset construction for least square regression. The paper is also well-written. I will fight for accepting this paper. Minor Comments 1. It might be worth mentioning that a coreset of size Omega(d^2) is in fact the best one can hope for if the covariance matrix needs to be preserved exactly. This can be seen when A is the edge-vertex incidence matrix of a complete graph. 2. It might be better to add more details about Algorithm 8 and provide a full proof for readers who are not familiar with Caratheodory's Theorem. 3. Line 77: which lower bound in [13] are you referring to?

Reviewer 3



Originality - The idea of evolving core-sets in order to improve running time of linear solvers is new to me. The paper introduce the idea of core-sets, which by the review has been studied before, but on relatively small scales. The work boost the performance of existing linear solvers using new methods Clarity - The submission is very well written and organized. there are some typo's (lines 47 , 68 , 84 , 185 ) some of them are important for the consistency of the math in the paper. The review and the motivation is clear and helpful. proofs are well written. The paper provides enough details to reproduce the results. Section 1.3 is exceptional. It suppose to give a good intuition for the framework but this part is not formal, and harder to understand. e.g. line 117 "a pair of coreset and sketch construction scheme" did not understand what type of input to expect , line 119 "faster", compare to what? it was unclear. Section 2 made everything clear and well defined. So maybe section 1.3 could be left out. Quality - Claims are supported. The paper misses mentioning the ratio needed between sample size n , a sample dimension d, and parameter k, so the reader could completely understand the running time of algorithm 2. Empirical evaluation demonstrate the practical potential of their method. Significant - This work has practical importance. practitioners are likely to easily implement suggested method to improve existing tools. Theoretical aspect could be motivated to other boosting method for improving running time for linear solvers. Or investigate the importance of Coresets for other use. === After reading authors feedback and other reviews indicating this work is first to offer an accurate solution to the problem and a, I tend to put even higher score on this paper. The paper has great significant. Clear accept.

[Author Response · NeurIPS 2019]

We thank the reviewers for the supportive scoring and very helpful suggestions.

The typos were corrected with your suggestions above, and the new version was sent to a professional English Editor.
Below are detailed answers.

**Reviewer 1:**

• Q: To the best of my knowledge, the paper is the first to introduce an *exact* coreset/sketch construction for
linear regression problems.
A: Indeed. We thank the reviewer for emphasizing this in the review.

• Q: "ordered finite set" is strange. A tuple notation $(p_1, ..p_n)$ is clearer.
Q: "How about "rows of C ... are linear combinations"
Q: "It's not a weighted subset. Its rows coincide with a certain subset of rows"
Q: "it seems [27] explains Theorem 2.2 as well"
Q: "that can be computed.." –> "that is computed.."
Q: The explicit remark in line 141 seems unnecessary
Q: The font size used in the figures in Figure 2 are way too tiny.
A: We thank the reviewer for the useful suggestions. We accepted all of them and updated the notation
accordingly.

• Q: "linear in the input $O(nd)$ for asymptotically large $n$" is a weird way of ignoring the dependency on $d$
A: It was changed to the precise computation time, as in the main claim.

• Q: Provide actual code to reproduce the results.
A: As promised, full open code will be published upon acceptance. It can also be sent now to the reviewers if
necessary.

**Reviewer 2:**

• Q: This is a groundbreaking result in the area of sketch and coreset. The new algorithm is elegant yet very
interesting. I will fight for accepting this paper.
A: We thank the reviewer for the high scoring and the supportive feedback.

• Q: It might be worth mentioning that a coreset of size $\Omega(d^2)$ is in fact the best one can hope for.
A: Indeed. A formal lower bound was added following this comment.

• Q: Add full proof for Caratheodorys Theorem and more details to Algorithm 8.
A: Added full proof and intuition.

• Q: which lower bound in [13] are you referring to?
A: This was a typo. We referred to the paper "Low Rank Approximation Lower Bounds in Row-Update
Streams" by D. Woodruf.

**Reviewer 3:**

• Q: The submission is very well written and organized. The review and motivation is clear and helpful. Proofs
are well written. This work has practical importance. Practitioners are likely to easily implement suggested
method to improve existing tools.
A: We thank the reviewer for the supportive review.

• Q: "Section 1.3 is not formal. I Did not understand what type of input to expect"
A: Section 1.3 was re-written per reviewer suggestion.

• Q: How the parameter $k$ affects the accuracy of Algorithm 2.
A: As stated, the algorithm is accurate for every value of $k$ in theory. Indeed, in practice, the numerical error is
smaller for high values of $k$. This is because in this case the size of each sketch is smaller.

• Q: Add analysis to the optimal value of $k$.
A: An interesting analysis was added. We thank the reviewer for this good suggestion.

• Q: Few Typos were found.
A: We thank the reviewer for the very careful reading and fixed the mentioned typos.

[Meta-Review · NeurIPS 2019]

This is an excellent submission, where a novel and fast coreset construction method with no approximation error for least square regression is proposed. It can be used to build more efficient and faster least square solvers. The paper is significant as it discusses important theoretical results, which have a high practical relevance. Congratulations to the authors on an excellent submission.